# Suppressing Optical Losses in Solar Cells via Multifunctional and Large-Scale Geometric Arrays

**DOI:** 10.3390/nano13202766

**Published:** 2023-10-16

**Authors:** Xiangqian Shen, Sihan Jiang, Xiaodan Wang, Hua Zhou, Zhiqiang Yu

**Affiliations:** 1Xinjiang Key Laboratory of Solid State Physics and Devices, School of Physical Science and Technology, Xinjiang University, Urumqi 830046, China; sxqlyq@xju.edu.cn (X.S.); jiangsihan2023@sina.com (S.J.); 2State Key Laboratory of Metal Matrix Composites, Shanghai Jiao Tong University, Shanghai 200240, China; 3School of Physics, Shandong University, Jinan 250100, China; 19820190154692@stu.xmu.edu.cn (X.W.); zhouhua2018@sdu.edu.cn (H.Z.); 4School of Electronic Engineering, Guangxi University of Science and Technology, Liuzhou 545006, China; 5Wuhan National Laboratory for Optoelectronics, School of Optical and Electronic Information, Huazhong University of Science and Technology, Wuhan 430074, China

**Keywords:** solar cells, light harvesting, geometric arrays, photonic structures, optical properties

## Abstract

The occurrence of optical loss on the surface of solar cells is inevitable due to the difference in the refractive index between air and glass, as well as the insufficient absorption of the active layer. To address this challenge, micron-sized geometry arrays, such as hemispheres and hemisphere pits, are prepared on quartz glass through the advanced indirect patterning technology of UV-LIGA. These geometric arrays exhibit multiple mechanisms for controlling light waves, including multiple rebounds, diffraction scattering, and total internal reflection. These synergistic effects suppress optical losses at the device’s surface and prolong the photon propagation path in the active layer. After being patterned with this structure, the average transmittance and haze of the quartz glass reach 93.91% and 75%, respectively. Compared to their flat counterpart, the decorated monocrystalline silicon solar cells demonstrated an apparent improvement in photocurrent and produced a 7.2% enhancement in power conversion efficiency.

## 1. Introduction

Energy scarcity and ecological contamination have always been persistent concerns for humanity. Driven by the pursuit of sustainable energy options, significant resources have been devoted to unlocking the vast benefits of solar power, characterized by its environmentally sound and renewable attributes as well as its cost-effective potential [1,2,3]. Among the multitude of technologies used to harness solar energy, solar cells based on the photovoltaic effect stand out prominently. This type of device absorbs solar radiation and directly converts it into electrical energy. Over the past several decades, there has been a remarkable surge in the advancement of solar cell technology. Presently, a diverse range of solar cell types has emerged, including crystalline silicon, silicon-based thin films, gallium arsenide, copper indium gallium selenide, dye-sensitized solar cells, and perovskite solar cells [4,5,6,7]. With its clean and sustainable attributes, solar cells are becoming a formidable challenge to the dominance of traditional fossil fuels. To claim victory in this competition, it is imperative to enhance the power conversion efficiency of solar cells, which relies on purifying materials, intricate electrical contact, and effective light management [8,9,10,11]. In particular, reducing reflection loss and enhancing the optical path in active materials are the primary principles for the design of solar cells with high performance [12]. Various strategies, such as sub-wavelength periodic structures, metallic plasmonics, and photonic crystal structures, have been demonstrated to play critical roles in light harvesting [13,14,15]. Despite the various advancements in solar cell architecture, the air/glass surface remains a persistent challenge as it entails an inevitable loss of ~4% due to the difference in refractive indices, as calculated by the Fresnel formula. Moreover, the presence of metal electrodes in the cell structure contributes to further optical loss as it reflects a fraction of unabsorbed light back to the glass. For instance, in the widely used monocrystalline silicon solar cell, a flat glass surface reflects an average of ~10–15% of the light in the 400–1100 nm wavelength range, which could be absorbed by silicon to generate a photocurrent [16]. Given this, reducing optical losses at the air/glass surface represents a significant opportunity to enhance the solar cell’s power conversion efficiency.

Although dielectric thin films like Si_3_N_4_, MgF_2_, with an optical thickness equal to one-quarter of the wavelength, are effective at reducing reflection at specific wavelengths and angles of incidence [9,12], these single-layer coatings fall short when meeting the needs of solar cells. This is primarily due to the fact that solar cells have a broad absorption spectrum ranging from visible light to near-infrared light. In order to encompass a wider range of responsive wavelengths, many approaches have been proposed. Some of these include porous glass [17], biomimetic moth-eye structures [18], randomly arranged pyramid structures [19], hemisphere pits [20], and gold square-shaped nanopillar arrays [21]. These methods enable the manipulation of light waves through gradient refractive indices, multiple rebounds, or plasmonic resonances, resulting in broadband, omnidirectional anti-reflection effects. Regrettably, the size and morphology of the patterns reported thus far fall short of achieving the desired results due to the inherent limitations of conventional preparation methods. For instance, pyramids serve as a prevalent anti-reflection structure. This structure is initially etched on the crystalline silicon surface employing alkaline solutions, such as NaOH, followed by subsequent transfer onto the targeted glass surface utilizing techniques like nanoimprinting. Nevertheless, the dimensions of the pyramids obtained through this method exhibit a random distribution ranging from 5 to 10 μm, while the inclination angle of their faces remains fixed at 54.7 degrees [22]. To enable multiple refractions of incident light within the pyramid, it is necessary for the pyramid dimensions to greatly surpass the wavelength of incoming light. Taking into account the fact that the response cut-off wavelength of crystalline silicon cells is approximately 1.1 μm, the ideal dimensions of the pyramid structure would need to exceed 10 μm [19]. Consequently, the optical design of cell devices has not yet been perfected in accordance with the original intent.

In this report, we present an alternate approach to UV-LIGA, which enables the development of geometric arrays at air/glass interfaces. Here, UV refers to ultraviolet, while LIGA is the German acronym for “Lithographie, Galvanoformung, Abformungto”, which denotes the process of etching, electroforming, and duplicating micro-scale patterns [23]. The standout advantage of UV-LIGA lies in its capability to fabricate a diverse range of geometric structures while allowing for precise control over their dimensions, periodicity, arrangement, and morphology, tailored to meet specific requirements. Herein, we demonstrate the utilization of the UV-LIGA technique to fabricate arrays of micron-sized hemispheres and hemispherical pits. The experimental and simulated results affirm that these large-scale geometric arrays have the ability to regulate the propagation of light waves at the air/glass interface from various perspectives, including multiple rebounds, diffraction scattering, and total internal reflection. With this particular design, the transmittance and haze of quartz glass are significantly enhanced. As a result, the surface optical losses of corresponding crystalline silicon solar cells were drastically suppressed, leading to a notable increase in their photocurrent density and power conversion efficiency.

## 2. Preparation of Geometric Arrays via UV-LIGA

For this experiment, the substrate of choice was quartz glass. The process of fabrication is illustrated in Figure 1, wherein the most pivotal steps encompass lithography, thermal reflux, micro-electroforming, and nanoimprint. Prior to preparation, quartz glass needs to undergo a cleaning process to eliminate impurities and oil stains on its surface. The sample is first immersed sequentially in acetone, ethanol, and deionized water, followed by ultrasonic treatment for 10 min each. Subsequently, the sample is dried using a nitrogen gun. To remove surface moisture and enhance adhesion, the dried sample requires further thermal treatment. It is subjected to a heat drying process in an oven at a temperature of 120 °C for a duration of 30 min.

(1) Lithography. The choice of photoresist plays a vital role in determining the outcome of an experiment. In this study, a commercially available positive photoresist (EPG590, Everlight Chemical, Taipei) was employed. First, a pristine layer of photoresist was spin-coated onto the immaculately cleansed surface of the quartz glass. This process was initiated with a gentle rotation speed of 500 r min^−1^ for a duration of 5 s, which was then swiftly followed by an elevated velocity of 1000 r min^−1^ for a remarkable period of 30 s. As a result of this meticulous process, the photoresist film achieved a thickness of ~11 μm. Subsequently, the sample underwent a pre-baking process on a heated stage. This step aimed to eliminate any excess water and solvents, as well as solidify the photoresist. The pre-baking temperature was set at 100 °C, with a duration of precisely 3 min. The aperture of the photoresist mask was expertly etched to achieve a diameter and period of 10 μm and 12 μm, respectively. The exposure procedure employed a contact mode with an exposure energy of 60 mJ cm^−2^. For the development step, a solution comprising NaOH with a concentration of 5% was selected as the developer, with a development time of 30 s. Following the photolithography process, a cylindrical array with a diameter of 10 μm and a periodicity of 12 μm was achieved on the surface of the quartz glass, as depicted in Figure 1c.

(2) Thermal reflux. The temperature and duration of the thermal reflux process were set at 150 °C and 3 min, respectively. Under the influence of high temperature, the cylindrical morphology of the polymer material transitioned into a partially molten state. Driven by surface tension, these cylindrical entities gradually diffused and transformed into hemisphere-shaped geometries, as illustrated in Figure 1d. In addition, this process effectively reduced the surface roughness of the microstructures.

(3) Micro-electroforming. First, a 100 nm-thick layer of nickel was thermally evaporated onto hemisphere arrays to serve as a seed for electroforming. The electroforming process entailed the microstructure to be utilized as the core mold (cathode) and nickel as the electroplating material (anode). The composition of one liter of the plating solution is as follows: Ni(NH_2_SO_3_)_2_·4H_2_O (450 mL), NiC_l2_·6H_2_O (5 g), H_3_BO_3_ (35 g), LN-MU (7 mL), LN-MA (1 mL), and surface additive (6 mL). The plating bath was maintained at a pH of 4 and heated to 47 degrees. A current density of 1 A dm^−2^ was applied to the cathode, while the anode’s current density was set to 2 A dm^−2^. The metal ion migrated toward the cathode due to the electrical potential difference between these two electrodes. Consequently, the ion grew into hemisphere structures that were consistent in shape with the cathode. Following the stripping process, a layer of nickel with a shape that was inverse to the hemisphere, namely a hemispherical pit, was obtained, as depicted in Figure 1g.

(4) Nanoimprint. To commence this process, a layer of ultraviolet nanoimprint resistance was applied onto the target quartz glass through spin coating. Subsequently, the prepared nickel template was superimposed onto the imprinting layer. Through the application of precise heat and pressure, the imprinting layer was diligently prompted to saturate every crevice of the hemispherical pits within the nickel template. Following this, the UV-exposure-induced curing process was conducted on the imprinting adhesive, with the light source energy set at 20 mJ cm^−2^. Finally, through the demolding technique, the nickel template and imprinting adhesive were gently separated, resulting in the formation of hemispherical arrays on the surface of the quartz glass, as depicted in Figure 1(h1). By employing the hemispherical arrays as a template and repeating the aforementioned imprinting steps, a quartz glass with hemispherical pit arrays was obtained, as illustrated in Figure 1(h2).

## 3. Characterization and Calculation

The surface morphologies of the geometric arrays were characterized through the utilization of scanning electron microscopy (SEM, Thermo Fisher Scientific, Waltham, MA, USA). The wetting properties of the sample surface were characterized using a contact angle measurement instrument (DSA100, Hamburg, Germany), with the solution employed being water. The transmission, reflection, and haze of the samples were measured using the UV-VIS spectrophotometer (UV-2600, Shimadzu, Kyoto, Japan). The current–voltage characteristics (I-V curve) of the solar cells were measured using a solar simulator (WXS-90S-L2, AM 1.5 GMM, Tokyo, Japan). During the testing process, the light source maintained an irradiance intensity of 100 mW m^−2^, while the temperature and relative humidity were carefully regulated at 25 °C and 50%, respectively.

The propagation and distribution of photons within geometric arrays were calculated using a three-dimensional finite difference time domain (FDTD) method. During modeling, the morphology, dimensions, periodicity, and arrangement of the geometric arrays were maintained consistently with experimental measurements. The spatial grid was discretized into cubic cells with a side length of 5 nm, while the time step was set to 0.4 ns. The light source employed a monochromatic wave emitted from the air, which vertically irradiated the surface of geometric arrays. The boundary conditions were established as follows: the direction of light propagation was defined by a perfectly matched layer (PML), while the direction of normal incident light adopted periodic boundary conditions (PBC).

## 4. Results and Discussion

Figure 1(i1) and (i2) illustrate the SEM images of the as-prepared hemisphere and hemispherical pit arrays, respectively. It can be seen that these geometric arrays were arranged in a regular and orderly square formation, with precise control over both shapes and sizes. The diameter of each hemisphere/hemispherical pit measured 10 μm, with a periodicity of 12 μm and a height/depth of 5 μm. Upon scrutinizing the magnified image, one could discern that the contours of the hemisphere/hemispherical pit were smooth, well-defined, and free from any signs of distortion or deformation. In order to elucidate the modulation effects of these geometries on incident light, we characterized their transmittance spectrum and haze levels, as depicted in Figure 2. Here, their haze is defined as the ratio of the diffuse transmission to the total transmission.

Evidently, the glass featuring micron-sized geometry arrays remarkably improved the transmission of visible wavelengths and slightly decreased it in the near-infrared region compared to the flat sample. The average transmission in the visible spectrum rose from 90.51% in the flat sample to 93.12% and 93.91% in the hemisphere and hemisphere pit, respectively. Interestingly, the haze increased from an initial near-zero value of 4.01% to about 75% across all measured bands. This elevated transmission and haze signified a greater number of photons traversing the air/glass interface, with the majority being diffused.

For short- and medium-wave photons, the incident rays underwent multiple reflections within the micro-sized structures due to the feature size of the geometry being much larger than the wavelengths of the photons. Based on Snell’s law, the reflectance of a photon passing through the air/glass interface once is R=4%. This value then decreases to $R^{n}$ after undergoing n rebounds within the micro-sized structures. Using this principle, one could calculate the transmittance $T_{1}$ of incident light at the geometric air/glass interface by implementing the formula $T_{1}=100\%-\sum_{1}^{n}p_{n}R^{n}$. Here, $p_{n}$ represents the proportion of photons undergoing n instances of rebound in relation to the total incident photons; that is, $\sum_{1}^{n}p_{n}=1$. Consequently, considering the existence of another flat glass/air interface, the total transmittance $T_{2}$ of light passing through a glass is $T_{2}=(1-R)(100\%-\sum_{1}^{n}p_{n}R^{n})$. In the event that all incident photons experience a sufficient number of rebounds, the part of $\sum_{1}^{n}p_{n}R^{n}$ approaches 0. Therefore, the limiting values of $T_{1}$ and $T_{2}$ for glass with one side patterned are 100% and 96%, respectively. However, as illustrated in the inset of Figure 2a, the transmittance of both the hemisphere and hemisphere pit were overtly below the prescribed limit of 96%. This deviation is suspected to be attributed to the absorption and scattering of impurities in the glass. In order to exclude the influence of the glass itself on this discrepancy, Figure 2b displays the difference in transmission, represented by ΔT, between the micro-sized geometry and flat glass. Based on the aforementioned analysis, the limit value of ΔT is 4%. As depicted in Figure 2b, the ΔT of the hemisphere and hemisphere pit first increased and then decreased, exhibiting a considerable improvement in the visible spectrum compared to the flat sample. More promisingly, the ΔT value approached or even exceeded the limit value in the wavelength range of 450–550 nm. This implies that the enhancement of transmission is not solely attributed to multiple rebounds but is also influenced by other factors, such as diffraction scattering and total internal reflection. As the wavelength increases, the diffraction due to grating becomes increasingly prominent. Fortunately, diffraction scattering alters the angle of incidence of photons, as evidenced by the haze curves in Figure 2a. Consequently, the optical path of photons in the active layer is prolonged.

Figure 3 illustrates the working principles of different anti-reflective structures. As illustrated in Figure 3a, for planar-structured solar cells, significant reflection losses occurred on the cell surface, where the optical path length of photons within the cell equaled the thickness of the absorbing layer. As depicted in Figure 3b,c, although the layered anti-reflective coating and subwavelength periodic structure could partially alleviate surface reflection losses, they did not alter the path of photon propagation. By contrast, the presence of a large-scale geometric array allowed for the versatile manipulation of light waves, concurrently suppressing surface optical losses and prolonging the propagation path of photons within the responsive layer, as illustrated in Figure 3d,e. To monitor and analyze the behavior of photons within micro-sized geometries and visually map their distribution at the air/glass interface, a three-dimensional electromagnetic simulation model was constructed and implemented via the FDTD method. Figure 4(a1–c1) and Figure 4(a2–c2) depict the sectional and top views, respectively, of the spatial electric field distribution resulting from the incidence of light with different wavelengths. In accordance with Maxwell’s electromagnetic field theory, the energy of an optical wave primarily depends on its electric field component. Consequently, the distribution of the electric field within a geometric structure indirectly reflects the distribution of photons within it.

In the scenario where λ=400 nm, the micro-sized geometries exhibited multiple rebounds as the ray traveled toward the air/glass interface. This phenomenon can clearly be observed within the area demarcated by the white circle in Figure 4(a1). Compared to the size of the hemisphere, the optical wave at this moment manifested its particle-like characteristics entirely. Multiple rebounds resulted in the uniform dispersal of energy in the optical wave within the hemisphere. Additionally, interference fringes were not observed in the air. This observation suggests that after multiple rebounds, the incident light and reflected light were no longer confined to the same plane, and the path difference between them was no longer constant. As the wavelength of the incident light increased, the particle-like characteristics of photons within the hemisphere diminished while their wave-like nature progressively strengthened. At a wavelength of 600 nm, one can concurrently observe the phenomena of multiple rebounds and the diffraction scattering of photons within the hemisphere, as illustrated in Figure 4(b1). However, during this instance, multiple rebounds were noticeably weaker compared to when *λ* = 400 nm, and one could observe interference fringes between the incident and reflected light in the air. As the incident light wavelength increased to 1000 nm, the phenomenon of multiple rebounds dissipated and was replaced by conspicuous diffraction scattering, as illustrated in Figure 4(c1). Simultaneously, the interference fringes in the air became remarkably pronounced. The corresponding top view of these three scenarios offers additional evidence supporting these phenomena from another perspective. As illustrated in Figure 4(a2), the distribution of the optical wave was uniform, owing to the unordered and angle-independent nature of the rebound in geometry. By contrast, the presence of diffraction is evident in Figure 4(c2), as indicated by the concentrated light spot. Figure 4(a3–c3) displays the transmitted diffraction efficiency vs. the order. It can be clearly observed that with the increase in wavelength, direct transmission (0 order) was significantly suppressed, while diffraction gradually strengthened, and the transmitted power tended toward higher-order diffraction, indicating that the transmitted light had a high diffraction angle. These observations are in good agreement with the aforementioned experimental results, providing additional evidence for the exceptional light-controlling properties exhibited by these geometric arrays.

Building upon this aforementioned understanding, these geometries were then implemented onto the surfaces of single-crystalline silicon solar cells. Subsequently, we conducted measurements on the reflection spectrum and current–voltage characteristics of these devices. As illustrated in Figure 5a, the reflection of solar cells demonstrated a noticeable decrease under the modulation of geometric arrays, signifying the absorption of a greater number of photons. For the purpose of a more comprehensive comparison, we calculated the weighted average reflectance $R_{av}$ via the following equation:(1)$$R_{av}=\frac{\int_{\lambda_{min}}^{\lambda_{max}}S(\lambda )R(\lambda )d\lambda }{\int_{\lambda_{min}}^{\lambda_{max}}S(\lambda )d\lambda }$$
where S(λ) denotes the standard AM1.5 solar spectrum, R(λ) represents the reflectance spectrum obtained from the measurements, and $\lambda_{min}$ and $\lambda_{max}$ are, respectively, the minimum and maximum response wavelengths of the cell devices. For single-crystalline silicon solar cells, the values for $\lambda_{min}$ and $\lambda_{max}$ corresponded to 400 nm and 1100 nm, respectively. The calculated results are shown below: $R_{av}^{flat}=15.14\%,\;{\;R}_{av}^{hemi}=7.26\%,\;{\;R}_{av}^{hemi\;pit}=7.27\%$. Evidently, the difference between the average reflectance values, denoted by ${\Delta R}_{av}$, for the geometries and flat surfaces, exceeded 4%, highlighting the impact of light trapping in the cell devices. These synergistic effects led to a significant enhancement in the photo-generated current. As illustrated in Figure 5b, it is evident that the short-circuit current density demonstrated a notable increase from 37.22 mA cm^−2^ for the flat surface to 39.83 and 39.87 mA cm^−2^ for the hemisphere and hemisphere pit structures, respectively. Furthermore, it is noteworthy that the introduction of geometric arrays, as opposed to a planar structure, does not alter the open-circuit voltage and fill factor of solar cells. This elucidates the fact that this optical modulation strategy does not undermine the inherent structure and electrical contacts of the devices. As a consequence of the increase in the photocurrent, the photoelectric conversion efficiency corresponding to the hemisphere and hemisphere pit structures augmented from 18.76% to 20.09% and 20.11%, respectively. This marks an impressive enhancement of approximately 7.2% compared to the planar one.

Finally, the exposure of photovoltaic components to outdoor conditions rendered them susceptible to the detrimental effects of rainwater erosion and dust accumulation, which significantly impacted their stability and efficiency. Enhancing surface hydrophobicity stands as one of the approaches to address these issues. Taking these considerations into account, we further characterized the surface wettability of these geometric arrays. As illustrated in Figure 6, the contact angle of water on the flat glass measured 28.8 degrees. However, after applying a flat adhesive, hemisphere, and hemisphere pit, this value surged to 77.4, 108.7, and 112 degrees, respectively. This demonstrates that the incorporation of geometric arrays enhances the hydrophobic nature of air/glass interfaces, ensuring exceptional self-cleaning properties. These traits allow for a reduction in contamination from dust and dirt, ultimately enhancing the efficiency of practical energy conversion when choosing a solar cell for outdoor applications.

## 5. Conclusions

In summary, our study demonstrates the production, replication, and transfer of micro-scale geometries at air/glass interfaces through UV-LIGA technology, effectively overcoming the inherent obstacle of direct patterning on glass surfaces. Distinct from traditional anti-reflective coatings and sub-wavelength structures, this sizable geometry affords numerous rebounds for short-wavelength light, while its periodic configuration facilitates robust diffraction scattering for long-wavelength light. As a result of these synergistic effects, there has been a significant reduction in optical loss at the air/glass interface and a considerable increase in haze. Encouragingly, the implementation of this design on silicon solar cells led to an increase in the photocurrent from 37.22 to 39.87 mA cm^−2^, resulting in an improvement in power conversion efficiency from 18.76% to 20.11%. Moreover, these geometric arrays offer other benefits, such as reusability, self-cleaning capabilities, and spectral independence, which enhance their potential for outdoor photovoltaic applications.

## Figures and Tables

**Figure 1 nanomaterials-13-02766-f001:**
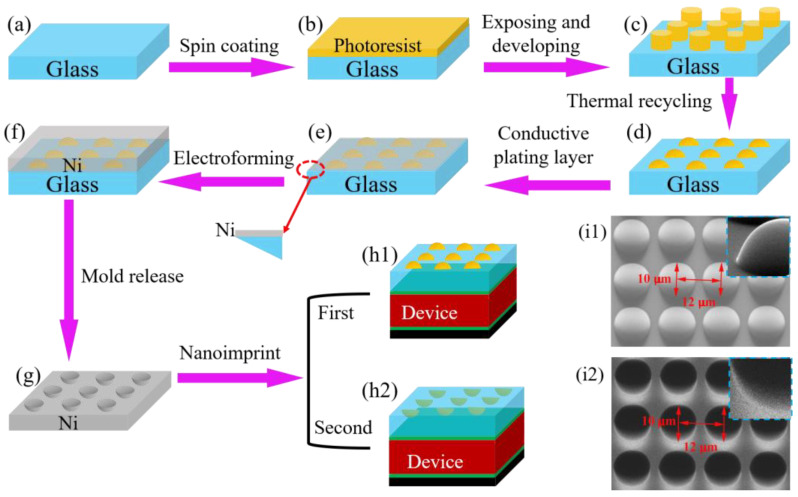
The diagrammatical representation and SEM images of micro-scale geometric arrays fabricated using the UV-LIGA process. (**a**–**c**) Lithography, (**d**) Thermal reflux, (**e**–**g**) Micro-electroforming, (**h1**,**h2**) Nanoimprint. SEM images of the (**i1**) Hemisphere and (**i2**) Hemisphere pit. The insets in (**i1**,**i2**) depict a magnified view of a quarter hemisphere and a quarter hemisphere pit, respectively.

**Figure 2 nanomaterials-13-02766-f002:**
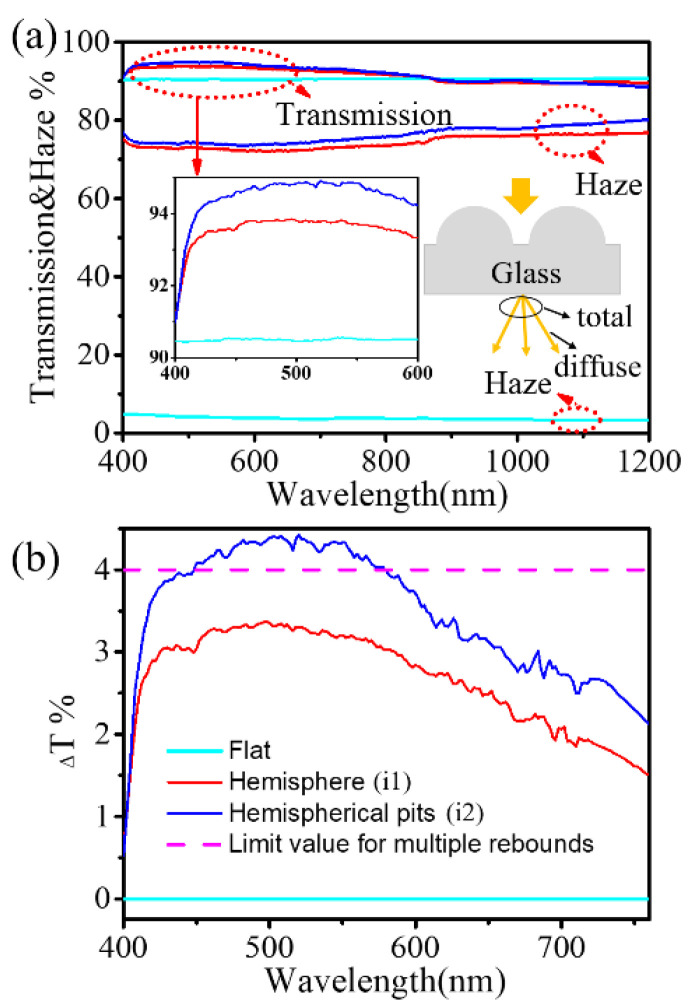
Optical characteristics of quartz glass coated with different shapes of geometries. (**a**) Transmission and haze, (**b**) The difference in light transmission between the geometric arrays and the flat glass.

**Figure 3 nanomaterials-13-02766-f003:**
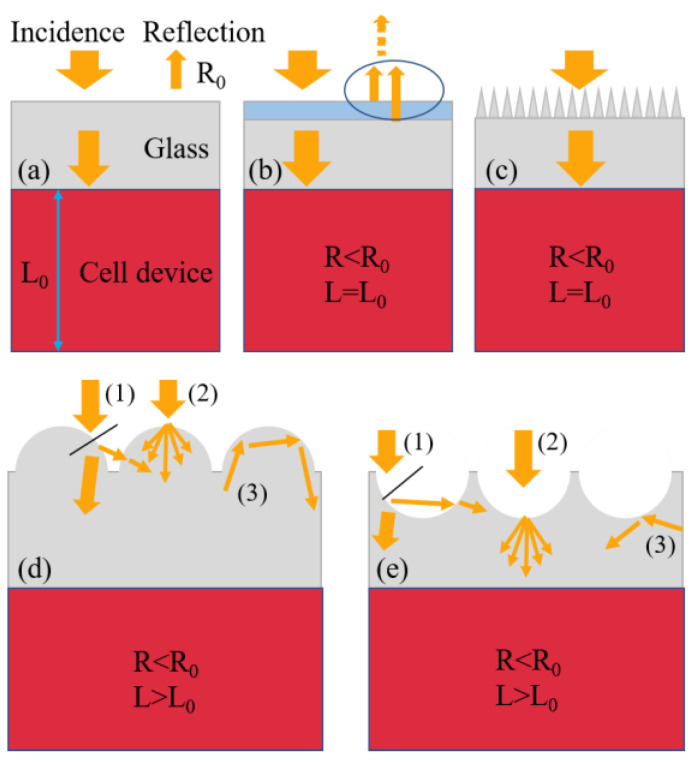
Working principle of various anti-reflective structures: (**a**) Solar cell without anti-reflective structure, (**b**) With layered anti-reflective coatings (destructive interferences), (**c**) With subwavelength periodic structure (gradient refractive index), (**d**,**e**) With micron-sized (**d**) Hemisphere and (**e**) Hemisphere pit arrays. (1), (2), and (3), respectively, symbolize the multiple rebounds, diffraction scattering, and total in-ternal reflection of light within geometric arrays. $R_{0}$ and $L_{0}$ denote the reflection and optical path length of photons within the cell, respectively, in the absence of any structured surface. When a structured surface with anti-reflective properties was present, R and L denote their respective values.

**Figure 4 nanomaterials-13-02766-f004:**
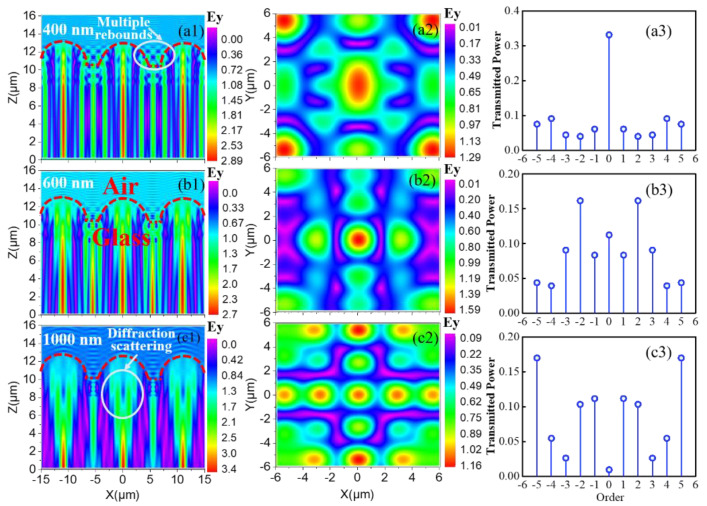
The cross-sectional view (**a1**–**c1**) and top view (**a2**–**c2**) of spatial electric field distribution at the air/glass interface for incident light with different wavelengths. (**a3**–**c3**) The transmitted diffraction efficiency vs. order.

**Figure 5 nanomaterials-13-02766-f005:**
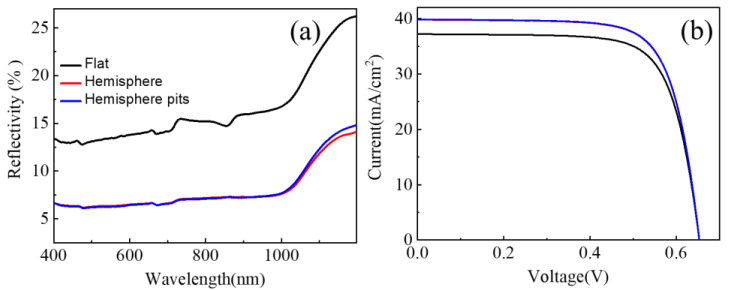
Reflection and current–voltage characteristics for cell devices coated with different shapes of geometries. (**a**) Reflection, (**b**) Current–voltage characteristics.

**Figure 6 nanomaterials-13-02766-f006:**
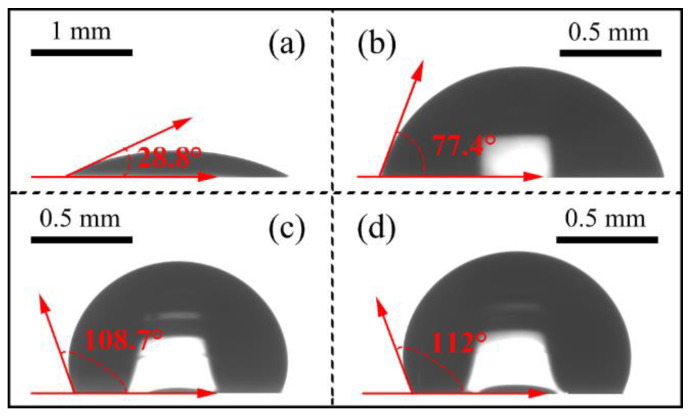
The contact angles of water at different air/glass surfaces. (**a**) Flat glass, (**b**) Flat adhesive, (**c**) Hemisphere, (**d**) Hemisphere pit.

## Data Availability

The data presented in this study are available on request from the corresponding author.

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
