# Peer review of "Suppressing Optical Losses in Solar Cells via Multifunctional and Large-Scale Geometric Arrays"

_nanomaterials, 2023, doi:10.3390/nano13202766_

Round 1
Reviewer 1 Report
The authors explored reducing optical loss in solar cells by designing hemispheres and hemisphere pits on quartz glass as micron-sized geometry arrays. This paper is well written, but the following revisions may be necessary.
1. Line 2 on page 2:
In the expression “400-1100 ?? wavelength”, the unit is written in italic. But the standard representation of a unit is not italic letters but roman as far as I know. Hence, the following revision may be necessary, but not mandatory.
?? -> nm (This revision may also be necessary in other parts.)
2. Line 4 on page 3
The authors wrote as “an ~11 ?? thick layer.” But, for the same reason as above, the following revision is necessary, but not mandatory.
?? -> ?m (This revision may also be necessary in other parts.)
Subsequently, ??, ?/??^2, and ????^−2 in other parts may also be revised in the same way.
3. Line 10 on page 2
The authors wrote as “To achieve this, three approaches have been proposed: porous glass, hemisphere pit arrays, and randomly arranged pyramid structures [17-19].” However, I think that the approaches along this line may be not three. There may be many different approaches in fact. For instance, “gold square-shaped nanopillars arrays” are suggested in Opt. Commun. 512, 128073 (2022).
I also suggest to cite other works in a way that:
porous glass, hemisphere pit arrays, and randomly arranged pyramid structures [X1,X2,X3]
-> porous glass [X1], hemisphere pit arrays [X2], and randomly arranged pyramid structures [X3]
In summary, I suggest to revise the sentence as “To achieve this, many approaches have been proposed: some of them are porous glass [X1], hemisphere pit arrays [X2], randomly arranged pyramid structures [X3], and gold square-shaped nanopillars arrays [X4].”
4. Line 14 on page 2
The authors wrote as “Regrettably, the sizes and morphologies of the patterns re-ported thus far fall short of achieving the desired results due to the inherent limitations of conventional preparation methods.” Regarding this, it may be good to write why conventional schemes are “fall short” especially in relation with the present research. For instance, the difference between the conventional hemisphere pit arrays and the present hemisphere pit arrays including why the present one is an alternate approach.
5. Typos: After Eq. 1on page 6
Where -> where
END
English is good.
Author Response
Dear Editors:
Thank you for your favorite consideration and for the reviewer’s comments concerning our manuscript entitled“Suppressing optical losses in solar cells via multifunctional and large-scale geometric arrays” (ID: nanomaterials-2617978). Those comments are valuable and very helpful for revising and improving our paper, as well as the important guiding significance to our researches. We have studied the comments carefully and have made correction which we hope meet with approval. In order to make the changes easily viewable for you and the reviewers, in the revised paper, we marked the revision with red color. The responses to the reviewer’s comments, point by point, are listed as following:
Respond to the reviewer’s comments:
- In the expression “400-1100 ?? wavelength”, the unit is written in italic. But the standard representation of a unit is not italic letters but roman as far as I know. Hence, the following revision may be necessary, but not mandatory.
Our responses:
Thank you for pointing out this question. We have conducted a thorough review of all unit notations in the text and have made the necessary corrections in the revised manuscript.
- The authors wrote as “an ~11 ?? thick layer.” But, for the same reason as above, the following revision is necessary, but not mandatory.
Our responses:
Thank you for bringing this issue to our attention, which has been corrected in the revised manuscript.
- The authors wrote as “To achieve this, three approaches have been proposed: porous glass, hemisphere pit arrays, and randomly arranged pyramid structures [17-19].” However, I think that the approaches along this line may be not three. There may be many different approaches in fact. For instance, “gold square-shaped nanopillars arrays” are suggested in Opt. Commun. 512, 128073 (2022). I also suggest to cite other works in a way that: porous glass, hemisphere pit arrays, and randomly arranged pyramid structures [X1,X2,X3]. In summary, I suggest to revise the sentence as “To achieve this, many approaches have been proposed: some of them are porous glass [X1], hemisphere pit arrays [X2], randomly arranged pyramid structures [X3], and gold square-shaped nanopillars arrays [X4].”
Our responses:
We acknowledge the reviewer’s comments and suggestions very much, which are valuable for improving the quality of our manuscript. Firstly, we diligently scrutinized the techniques employed in the production of micro/nano-patterns on glass surfaces in recent years, and incorporated relevant literature citations into the revised manuscript (Ref. [18, 21]). Secondly, in accordance with the suggestion provided by the reviewer, we have revised the phrasing in this section to ensure a harmonious correlation between the methods employed and the references cited.
- The authors wrote as “Regrettably, the sizes and morphologies of the patterns re-ported thus far fall short of achieving the desired results due to the inherent limitations of conventional preparation methods.” Regarding this, it may be good to write why conventional schemes are “fall short” especially in relation with the present research. For instance, the difference between the conventional hemisphere pit arrays and the present hemisphere pit arrays including why the present one is an alternate approach.
Our responses:
Thank you for your insightful comments. Based on the reviewer's feedback, in the revised manuscript, we have provided specific examples to illustrate the challenges encountered with traditional approaches in fabricating micro-nano structures. Additionally, we have emphasized the advantages of the method employed in this study.
- Typos: After Eq. 1on page 6
Our responses:
Thank you for pointing out this error, which has been corrected in the revised manuscript.
We appreciate for Editors and Reviewers’ warm work earnestly, and hope that the correction will meet with approval. Once again, thank you very much for your comments and suggestions.

Reviewer 2 Report
1. If the authors can show the cross-section view of (i1) & (i2) images, I think that it's better.
2. I understood that figure 2, however, as considered the readers, you must mark such as hemisphere (i1), hemispherical pits (i2) in Fig. 2(b).
3. If you want to show the working principle of various anti-reflective structures, you must show the structures such as flat, hemisphere and hemispherical pits.
4. Is this real data from experimental (fig. 5(b))?
5. In fig. 6, figure caption should be shown in below the figure.
6. I think that figure 6 is not necessary. The relationship between the contact angle and optical properties is not matched as I thought.
Overall you need to check English grammar.
Author Response
Dear Editors:
Thank you for your favorite consideration and for the reviewer’s comments concerning our manuscript entitled“Suppressing optical losses in solar cells via multifunctional and large-scale geometric arrays” (ID: nanomaterials-2617978). Those comments are valuable and very helpful for revising and improving our paper, as well as the important guiding significance to our researches. We have studied the comments carefully and have made correction which we hope meet with approval. In order to make the changes easily viewable for you and the reviewers, in the revised paper, we marked the revision with red color. The responses to the reviewer’s comments, point by point, are listed as following:
Respond to the reviewer’s comments:
- If the authors can show the cross-section view of (i1) & (i2) images, I think that it's better.
Our responses:
Thank you for your insightful comments. We attempted to provide cross-section images of the fabricated hemispheres and hemispherical pits. However, due to their polymer composition, these structures are prone to deformation during cutting, which does not accurately reflect their true contours. In order to ensure clarity for readers to visualize the profiles of these geometries, we adjusted the scanning angles. Furthermore, we magnified one-fourth of the hemisphere and one-fourth of the hemispherical pit (The insets in Fig. 1(i1) and Fig. 1(i2)). With these adjustments, the contours and boundaries of the geometries can be observed distinctly.
- I understood that figure 2, however, as considered the readers, you must mark such as hemisphere (i1), hemispherical pits (i2) in Fig. 2(b).
Our responses:
Thank you for bringing this issue to our attention. We have made the necessary modifications based on the reviewer's feedback
- If you want to show the working principle of various anti-reflective structures, you must show the structures such as flat, hemisphere and hemispherical pits.
Our responses:
Thank you for your insightful comments. Based on the reviewer's suggestions, we have made modifications to Figure 3 in the original manuscript by adding an illustrative diagram showcasing the propagation of light within the hemispherical pits. Additionally, we have provided a comparative analysis of the anti-reflective principles observed in different geometries.
- Is this real data from experimental (fig. 5(b))?
Our responses:
Yes, the I-V curves in Fig. 5(b) were derived from experimental measurements conducted using a solar simulator (WXS-90S-L2, AM 1.5 GMM, Japan). During the testing process, the light source maintained an irradiance intensity of 100 mW m-², while the temperature and relative humidity were carefully regulated at 25 degrees Celsius and 50%, respectively. In the revised manuscript, these experimental details have been added.
- In fig. 6, figure caption should be shown in below the figure.
Our responses:
Thank you for pointing out this error, which has been corrected in the revised manuscript.
- I think that figure 6 is not necessary. The relationship between the contact angle and optical properties is not matched as I thought.
Our responses:
Thank you for your insightful comments. We have given careful consideration to the reviewer's suggestions. Nevertheless, due to the following reasons, we believe it is necessary to retain Fig. 6. Firstly, the exposure of photovoltaic components to outdoor conditions renders them susceptible to the detrimental effects of rainwater erosion and dust accumulation, which significantly impact their stability and efficiency. Enhancing surface hydrophobicity stands as one of the approaches to address these issues. As shown in Fig. 6, the geometric array fabricated in this study demonstrates a larger water contact angle compared to the planar one, indicating superior hydrophobicity; Secondly, in addition to photovoltaic cells, these geometric arrays may also find applications in other photon-electronic devices such as sensors and detectors. Therefore, it is our intention to showcase to the readers the multifunctionality of this structure; Thirdly, we have revised certain statements in order to align the content depicted in Fig. 6 with the main theme of the article.
Finally, our manuscript has been meticulously revised based on the feedback from the reviewers. We have rephrased some sentences to enhance clarity and improve the flow of the text. Additionally, we have eliminated all grammar, usage, and spelling errors. Furthermore, a native English-speaking colleague has refined and polished the language of our manuscript.
We appreciate for Editors and Reviewers’ warm work earnestly, and hope that the correction will meet with approval. Once again, thank you very much for your comments and suggestions.

Round 2
Reviewer 1 Report
The authors revised the manuscript according to my report. Now I recommend the publication of it in Nanomaterials.